# Effects of Different Processing on miRNA and Protein in Small Extracellular Vesicles of Goat Dairy Products

**DOI:** 10.3390/nu16244331

**Published:** 2024-12-16

**Authors:** Yuqin Fan, Zhikang Li, Yanmei Hou, Chumin Tan, Sheng Xiong, Jinjing Zhong, Qiuling Xie

**Affiliations:** 1College of Life and Science Technology, Jinan University, Guangzhou 510632, China; 2Ausnutria Hyproca Nutrition Co., Ltd., Changsha 410200, China; 3Guangdong Province Key Laboratory of Bioengineering Medicine, Guangzhou 510632, China

**Keywords:** goat milk, small extracellular vesicles, dairy processing, miRNA, proteome

## Abstract

Objectives: Small extracellular vesicles (sEVs) are nanosized vesicles with biological activities that exist in milk, playing functional roles in immunity, gut balance, and the nervous system. Currently, little is known about the impact of processing on milk sEVs. Methods: In this study, sEVs were collected from raw goat milk (g-sEV), pasteurized goat milk (pg-sEV), and goat milk powder (*p*-sEV) using a sucrose cushion centrifugation combined with qEV chromatography. Then, the sEVs were identified and compared using NTA, Western blot, and TEM. After extracting RNA and the total proteome from sEVs derived from different samples, the RNA was subjected to high-throughput sequencing, and peptide fragments were analyzed using mass spectrometry. Finally, GO and KEGG pathway analyses were performed on the results. Results: The characterization results revealed a decrease in diameter as the level of processing increased. High-throughput sequencing results showed that all three types of small extracellular vesicles were found to be rich in miRNA, and no significant differences were observed in the most abundant sEV species. Comparing with g-sEV, there were 3938 and 4645 differentially expressed miRNAs in pg-sEV and *p*-sEV, respectively, with the majority of them (3837 and 3635) being downregulated. These differentially expressed miRNAs were found to affect biological processes or signaling pathways such as neurodevelopment, embryonic development, and transcription. Proteomic analysis showed that there were 339 differentially expressed proteins between g-sEV and pg-sEV, with 209 proteins being downregulated. Additionally, there were 425 differentially expressed proteins between g-sEV and *p*-sEV, with 293 proteins being downregulated. However, no significant differences were observed in the most abundant protein species among the three types of sEVs. Enrichment analysis indicated that the differentially expressed proteins were associated with inflammation, immunity, and other related processes. Conclusions: These results indicate that extracellular vesicles have a protective effect on their cargo, while processing steps can have an impact on the size and quantity of the sEVs. Furthermore, processing can also lead to the loss of immune-related miRNA and proteins in sEVs.

## 1. Introduction

Goat milk is one of the major dairy foods and is thought to have shorter gastric emptying time and better digestibility compared to cow milk [1]. In order to reduce the presence of pathogenic microorganisms and extend the shelf life of goat milk products, processing treatments such as homogenization, heat treatment, concentration, and spray drying are necessary before sale on the market. While these processing steps are essential, they can also have a negative impact on the nutritional components of goat milk. For example, homogenization reduces the lipid molecules in goat milk, and heat treatment affects the stability of protein, leading to protein denaturation and inactivation. Furthermore, the protein can react with lactose, altering the taste of the goat milk [2,3].

Small extracellular vesicles (sEVs) are small vesicles secreted by cells, with a diameter ranging from 30 to 200 nm [4,5]. These vesicles have a lipid bilayer membrane structure and contain proteins, lipids, nucleic acids, and other substances [6]. It has been reported that milk also contains a significant amount of sEVs. The lipid bilayer membrane of milk sEVs provides protection for their contents in the digestive environment and facilitates their absorption by cells [7,8,9,10]. Additionally, milk sEVs have the unique ability to penetrate the intestinal barrier and blood–brain barrier, playing a crucial role in immune regulation, inflammation prevention, and intestinal health [11]. Research demonstrated that fluorescently labeled milk sEVs, when orally administered to mice, can enter the bloodstream and be detected in various organs, including the liver, spleen, lung, and kidney [12]. In a study conducted by Zhou et al. [13], miRNA sequencing of human milk sEVs revealed a high abundance of immune-related miRNAs, suggesting their importance in infant immune system development. Furthermore, Admyre et al. [14] found that sEVs expressing CD86 and MHC II could stimulate CD4+ T cells and increase T cell numbers when added to peripheral blood mononuclear cells, indicating the immunomodulatory abilities of sEVs.

Although there is extensive research on human and bovine milk sEVs, limited studies have been conducted on goat milk sEVs, particularly regarding the effects of processing on their composition. Given the biological functions and contents of sEVs, it is worthwhile to explore the components of goat milk sEVs and their processed products. In this study, we employed sucrose cushion centrifugation and size exclusion chromatography to extract sEVs from raw goat milk, pasteurized goat milk, and goat milk powder. Subsequently, miRNA sequencing and proteomics were utilized to analyze and compare the sEVs from the three types of milk, aiming to investigate the impact of processing on goat milk sEVs, including their quantity, particle size, miRNA, and protein composition.

## 2. Materials and Methods

### 2.1. Acquisition and Pretreatment of Goat Milk Samples

Raw goat milk, pasteurized goat milk, and goat milk powder were produced and provided by Ausnutria Hyproca Nutrition Ltd. (Changsha, China) from Saanen goats—three replicates of each sample, all from different batches from the same company. Pasteurized milk was obtained by homogenizing raw milk and sterilizing it at 75–85 °C. Milk powder was obtained by concentrating and spray drying pasteurized milk. The raw goat milk and pasteurized milk were stored at −80 °C, while the goat milk powder was stored in a dry room at ambient temperature. To prepare the goat milk powder solution, 35 g of goat milk powder was dissolved in 200 mL of deionized water at a temperature of 45 °C. Raw goat milk and pasteurized milk were thawed at room temperature and placed in a centrifuge tube.

### 2.2. Extraction of Milk Small Extracellular Vesicles

First, the emulsion was centrifuged at 3000× *g* for 15 min at 4 °C to remove the upper fat, resulting in skim milk. Subsequently, the skimmed raw goat milk, pasteurized milk, and goat milk powder were pretreated with chymosin (TCI, Shanghai, China) at a concentration of 0.0002 g/mL. After 1 h, whey was obtained by centrifugation at 10,000× *g* for 30 min at 4 °C. To obtain the sEVs, 30% sucrose and whey were mixed and subjected to ultracentrifugation at 100,000× *g* for 70 min. The lower layer was then diluted with 1× PBS and centrifuged again at 100,000× *g* for 70 min. The resulting precipitate, which contains sEVs, was collected, re-suspended in PBS, and stored at −80 °C.

To further purify the re-suspended sEVs in whey, a size exclusion qEV column (IZON, Shanghai, China) was used. Specifically, 500 μL of the above-mentioned extracted sEVs were added to the qEV column, and 5 mL of 1× PBS, filtered through a 0.22 μm filter, was added to the column. The fractions were collected in EP tubes, with a total of ten fractions per 500 μL. Subsequently, the protein concentration of each fraction was measured using the Bicinchoninic Acid Assay kit (Thermofisher, Waltham, MA, USA).

### 2.3. Western Blot

Small extracellular vesicle samples of ten fractions were denatured, centrifuged, and electrophoresed in 12% SDS-PAGE gel and transferred to a nitrocellulose membrane. The membranes were blocked with tris-buffered saline with 0.1% Tween 20 (TBS-T) containing 5% skim milk, followed by incubation with CD63 (Proteintech, Wuhan, China; 1:3000 dilution) or TSG-101 primary antibody (Proteintech; 1:3000 dilution) overnight at 4 °C. Membranes were incubated with HRP-conjugated secondary antibodies (Proteintech; 1:5000 dilution) for 1 h at room temperature. Images were acquired using a Bio-Rad gel imaging system (Bio-Rad, Berkeley, CA, USA).

### 2.4. Nanoparticle Tracking Analysis (NTA)

Small extracellular vesicle samples were appropriately diluted with PBS, and then the diluted samples were injected into the nanoparticle tracking analyzer (NS300, Malvern, Tokyo, Japan). The trajectory of the small extracellular vesicles was recorded, and the particle size distribution and concentration of sEVs were obtained in Nano Sight NTA3.4.

### 2.5. Transmission Electron Microscope (TEM)

First, 10 μL of diluted small extracellular vesicle samples were added to the copper mesh, standing at room temperature for 3 min, then the excess samples were sucked away, and 10 μL of 5% phosphotungstic acid dye solution (Solarbio, Beijing, China) was added to stain for 1 min, and the excess dye solution sucked away. The staining was repeated three times, and finally washed three times with PBS, each time for 1 min. After the copper mesh was dried, it was placed under the TecnaiTM G2 Spirit transmission electron microscope (FEI, Hillsboro, OR, USA) to observe the morphology of the sEVs.

### 2.6. Extraction of miRNA, Small RNA Library Construction, and Sequencing

#### 2.6.1. Extraction and Detection of Milk Small Extracellular Vesicle miRNA

The miRNeasy Serum/Plasma Kit (Qiagen, Hongkong, China) was used to extract the small RNA from milk sEVs. The small extracellular vesicle sample was mixed with QIAzol lysate buffer in a 5:1 ratio. Chloroform, equal in volume to the sample, was added, followed by vortexing for 15 s and incubating at room temperature for 3 min. The mixture was then centrifuged at 12,000× *g* for 15 min at 4 °C. The upper aqueous phase was transferred to a new EP tube, and 5 μL of nucleic acid precipitating agent was added, followed by incubation at room temperature for 3 min. Anhydrous ethanol, 1.5 times the volume of the lysate, was added, and miRNA was collected by centrifugation using a RNeasy MinElute centrifuge tube at 8000× *g* for 15 s. Next, 700 μL of RWT solution was added, and after centrifugation at 8000× *g* for 15 s, the filtrate was discarded. This was followed by the addition of 500 μL of RPE solution, centrifugation at 8000× *g* for 15 s, and discarding of the filtrate. Subsequently, 500 μL of 80% ethanol was added, and the mixture was centrifuged at 8000× *g* for 2 min. The RNeasy MinElute centrifuge tube (Qiagen, Hongkong, China) was placed in a new 2 mL collection tube, the lid was opened, and centrifugation was performed at 12,000× *g* for 5 min to discard the filtrate. Finally, the RNA was eluted with an appropriate amount of RNase-free water, and the RNA concentration was measured using a Nanodrop (Thermo Fisher, Waltham, MA, USA). RNA integrity was assessed by electrophoresis using an Agilent 4200 (Agilent, Santa Clara, CA, USA) instrument.

#### 2.6.2. Small RNA Library Construction and miRNA Sequencing

The construction of the small RNA library and miRNA sequencing were performed by SHBio Biotechnology Co., Ltd. (Shanghai, China). The extracted RNA was purified using the RNA Clean XP Kit (Beckman Coulter, Brea, CA, USA, Cat.A63987) and treated with the RNase-Free DNase Set Kit (Qiagen, Hongkong, China, Cat.79254) to eliminate any contaminating DNA. After purification, the miRNA was ligated with the 3′ and 5′ end adaptors, and cDNA was generated through reverse transcription. Subsequently, amplification, enrichment, library size selection, and purification steps were performed. The generated library was then subjected to sequencing using the Illumina NovaSeq 6000 sequencer (Illumina, San Diego, CA, USA).

### 2.7. Mass Spectrum and Proteomic Analysis of Milk Small Extracellular Vesicle Proteins

#### 2.7.1. Extraction, Enzymatic Hydrolysis, and Peptide Acquisition of Milk Small Extracellular Vesicle Proteins

SEVs were lysed on ice with RIPA lysis buffer (containing 1% PMSF) (Beyotime, Shanghai, China) for 30 min, and 100 μg of lysed protein was mixed with DTT (Macklin, Shanghai, China) and incubated at 37 °C for 90 min. Then, 10 μL of IAA (1 mol/L) (Macklin) was added, and the protein was precipitated with acetone. The precipitated protein was re-dissolved and digested by trypsin (Macklin, Shanghai, China) at 37 °C for 16 h. After that, the peptides were concentrated by ultrafiltration and dried. A further desalination of the peptides was performed using a ZipTip column (Millipore, Burlington, VT, USA).

#### 2.7.2. Mass Spectrum Detection and Database Search

The peptides were re-dissolved in a 1‰ formic acid solution and mixed with iRT-Kit standard (Biognosys, Wagistrasse, Switzerland) before being pre-separated by HPLC using an Acclaim PepMap 100 C18 column (Thermo Scientific, Waltham, American) at a flow rate of 600 nL/min. The mobile phase A was a 1‰ formic acid solution, and the mobile phase B was an 80% acetonitrile solution containing 1‰ formic acid. The maximum injection time was set to 30 ms. The MS and MS/MS parameters were set as follows: (1) MS parameters: detector type—orbitrap; resolution—60,000 at 400 *m*/*z*; scan range—350–1200 *m*/*z*; AGC target—4.0 × 10^5^; maximum injection time—30 ms. (2) MS/MS parameters: isolation mode—quadrupole; activation type—HCD; collision energy—33%; detector type—orbitrap; orbitrap resolution—30,000 at 400 *m*/*z*; mass range—normal; scan range—200–2000 *m*/*z*; AGC target—5.0 × 10^5^; maximum injection time—50 ms.

### 2.8. Statistical Analysis

For analysis of miRNA sequencing results, the raw reads were screened and filtered using Fastx (version: 0.0.13) to exclude unqualified reads. Bowtie was used to compare the clean reads of 18–40 nt in length with the reference genome. miRNAs and other small RNAs were identified and annotated based on information from miRBase. The miRNA expression was normalized using the TMM method and converted into standardized miRNA expression per million transcripts (TPM). The statistical significance of enrichment was calculated using the Benjamini–Hochberg *p*-value adjustment method. All *p*-values were ranked in ascending order. For each ranked *p*-value p(i), a critical value of (i/m) × Q was calculated, where m is the total number of tests, i is the rank, and Q is the target FDR (0.05). The largest i for which p(i) ≤ (i/m) × Q holds was identified as significant. Adjusted *p*-values were then computed as min(1, max(j ≥ i)(p(j) × (m/j))) to ensure monotonicity and a maximum of 1. This method effectively identifies significant results while controlling the FDR. GO enrichment and KEGG pathway analyses were conducted using the DAVID database (https://david.ncifcrf.gov/, accessed on 15 April 2022), and the results were visualized using the R package ggplot2.

For mass spectrum and proteomic analysis, the raw results of mass spectrometry were further analyzed using Spectronaut 14.10 (Biognosys, Schlieren, Switzerland). The Biognosys Factory Settings (BGS) of directDIA were used to search the Fasta database of the corresponding species (UniProtKB 2021_04 results: Goat (TrEMBL 35398)). Protein identification and quantitative results were exported and analyzed against the DAVID database (https://david.ncifcrf.gov/) to perform functional enrichment of differentially expressed proteins, including GO and KEGG enrichment analysis.

## 3. Results

### 3.1. Extraction of sEVs from Goat Milk Samples

The sEVs from raw goat milk, pasteurized goat milk, and goat milk powder were extracted using sucrose cushion centrifugation combined with a qEV size exclusion column. To assess the presence of sEVs, Western blotting was performed to detect the levels of CD63 and TSG101, which are commonly used markers of sEVs. WB results showed that the eighth and ninth fractions of all three samples had higher levels of CD63 and TSG101, indicating the presence of sEVs (Figure 1A). In the Western blot (WB) results, both pg-sEV and *p*-sEV exhibited an increased number of CD63 bands compared to g-sEV, indicating that the processing procedure leads to an elevation in the number of CD63 bands. Notably, *p*-sEV displayed the highest number of CD63 bands, suggesting that as the degree of processing increases, CD63 may undergo further degradation. To further analyze the small extracellular vesicle content and protein concentration, nanoparticle tracking analysis (NTA) and BCA protein concentration analysis were performed. NTA measures the number of small extracellular vesicle particles, while BCA analysis quantifies the protein concentration of sEVs. The results showed that the eighth fraction had the highest small extracellular vesicle content and protein concentration among all the fractions for each sample (Figure 1B).

To assess the purity of the sEVs obtained through NTA and BCA analysis, the Webber and Clayton Methods were used [15]. The small extracellular vesicle purity was calculated as the ratio of small extracellular vesicle particle concentration to small extracellular vesicle protein concentration (NTA/BCA ratio, particles/μg). The results (Figure 1C) indicated that the eighth fraction had higher small extracellular vesicle purity compared to the other fractions for each sample. Based on the abundance of small extracellular vesicle content, high protein concentration, and high purity, the eighth fraction was selected for subsequent analysis.

### 3.2. Identification and Comparison of sEVs from Goat Milk Products

The raw goat milk sEVs (g-sEV), pasteurized goat milk sEVs (pg-sEV), and goat milk powder sEVs (*p*-sEV) obtained from three different types of goat milk products were further characterized.

To observe the morphology of the extracted sEVs from different milk products, transmission electron microscopy (TEM) was employed. The results (Figure 2A) demonstrated the presence of the characteristic cup-shaped structure of sEVs in all three samples. Nanoparticle tracking analysis (NTA) was conducted to determine the size of the extracted vesicles, which was found to be less than 200 nm in diameter (Figure 2B), consistent with the size range of sEVs. The particle sizes of g-sEV, pg-sEV, and *p*-sEV were measured to be 195.7 nm, 168.6 nm, and 143.0 nm, respectively. These findings indicated that the particle size of sEVs decreased with increase of processing steps.

Furthermore, the particle concentration of *p*-sEV (3.06 × 10^10^ particles) was significantly lower than that of g-sEV (1.77 × 10^13^ particles) and pg-sEV (3.34 × 10^13^ particles) (Figure 2C, Table 1), while there was no significant difference observed in the protein concentration (Figure 2D). In terms of small extracellular vesicle purity, *p*-sEV (3.98 × 10^9^ particles/μg) exhibited significantly lower purity compared to the sEVs derived from the other two dairy products (g-sEV: 8.34 × 10^9^ particles/μg, pg-sEV: 1.51 × 10^10^ particles/μg) (Figure 2E). In summary, the processing of raw milk into milk powder leads to a loss of milk sEVs.

### 3.3. Comparison of Small RNA in sEVs of Goat Milk Products

To investigate the impact of processing on small RNA, such as miRNA, in goat milk sEVs, we conducted small RNA extraction and sequencing analysis for g-sEV, pg-sEV, and *p*-sEV. The sequencing results (Figure 3A) indicated that the proportion of miRNA was highest in g-sEV (60%), followed by pg-sEV (52%), and *p*-sEV (25%). These findings suggest that processing could decrease miRNA in sEVs.

The length distribution of small RNA revealed that the majority of small RNA reads were 22 nt in length (Figure 3B), consistent with the length of miRNA. The 22 nt size class exhibited a high proportion of miRNA reads identified in the miRbase database, further confirming the presence of abundant miRNA in the extracted RNA. Notably, the reads value of g-sEV at 22 nt was higher than that of pg-sEV and *p*-sEV, and *p*-sEV also contained a substantial number of unannotated small RNAs. This observation further supports the notion that miRNA is lost during processing.

### 3.4. Comparison of miRNA Abundance in sEVs of Different Goat Milk Products

Venn diagram analysis was conducted based on the sequencing results to examine the identified miRNAs. As depicted in Figure 4A, a total of 458, 295, and 291 miRNAs were identified in g-sEV, pg-sEV, and *p*-sEV, respectively. Among these, 233 miRNAs were found to be shared, accounting for 51%, 79%, and 80% of the total miRNA species in g-sEV, pg-sEV, and *p*-sEV, respectively.

However, upon analyzing and comparing the top 10 miRNAs in the three sEVs, we observed little difference among three sEVs, particularly in g-sEV and pg-sEV. Three miRNAs in milk powder were different from the other two sEVs which still ranked in the top 10 miRNAs, indicating that sEVs have a certain protective effect on their contained miRNAs (Table 2).

Furthermore, hierarchical clustering analysis was performed on the miRNAs detected in each sample’s sequencing. The results demonstrated that, in the comparison between g-sEV and pg-sEV, only one newly predicted miRNA, novel.13, exhibited up-regulation in pg-sEV, while the expression levels of the other differentially expressed miRNAs were lower than those in g-sEV (Figure 4B). When comparing *p*-sEV with g-sEV in raw goat milk, the down-regulated miRNAs in *p*-sEV included miR-15b-5p, miR-150, and miR-345-5p, whereas the up-regulated miRNAs included miR-1 and miR-184 (Figure 4C). Generally, the differentially expressed miRNAs in *p*-sEV were predominantly down-regulated.

### 3.5. Function Analysis of the Top 10 Differentially Expressed miRNAs

To gain further insights into the impact of processing on milk sEVs, we performed GO enrichment analysis and KEGG pathway analysis on the target genes of the top ten differentially expressed miRNAs.

In the comparison between g-sEV and pg-sEV, the target genes of differentially expressed miRNAs were found to be involved in various biological processes, including the integrin-mediated signaling pathway. The cell components primarily enriched miRNAs related to nucleoplasm and cytosol. The molecular functions of the targeted genes are primarily related to binding activities, including ATP binding and chromatin binding (Figure 5A). As for the pathways in which the targeted genes are mainly involved, they include ECM-receptor interaction, focal adhesion, and relaxin signaling pathway (Figure 5B).

In the comparison between g-sEV and *p*-sEV, the most target genes of differentially expressed miRNAs were found to be involved in biological processes related to transcriptional regulation and DNA-templated processes. The cell components, primarily involving miRNAs related with, were nucleoplasm and cytosol. Differentially expressed miRNAs were enriched in the molecular functions related predominantly with binding and enzyme activities, including ATP binding and protein serine/threonine/tyrosine kinase activity (Figure 5C). Also, the pathways of the target genes that are mainly involved include focal adhesion, ErbB signaling pathway, and Axon guidance (Figure 5D).

These findings suggest that processing of milk sEVs can lead to differential regulation of miRNAs, which in turn affects various biological processes and pathways involved in cell signaling, adhesion, and development.

### 3.6. Comparison of Small Extracellular Vesicle Protein in Different Goat Milk Products

The proteins in the three sEVs were identified using mass spectrometry. Statistical comparison revealed that 910, 1174, and 798 proteins were identified in g-sEV, pg-sEV, and *p*-sEV, respectively. Among these, 612 proteins were shared by all three sEVs, accounting for 67%, 52%, and 77% of the total proteins in g-sEV, pg-sEV, and *p*-sEV, respectively (Figure 6A). However, the top 10 proteins in the three sEVs were mostly the same as each other. These proteins included functional proteins found in milk, such as BTN1A1, MFGE8, and XDH. Additionally, membrane-related proteins, such as CD36, MUC1, and CD9, were also present in the top10 proteins in sEVs, of which CD36 and CD9 are considered marker proteins of sEVs (Table 3).

After screening for differential proteins, a hierarchical clustering heat map was generated using the website (https://www.omicsolution.org/wkomics/main/, accessed on 15 March 2022). In the protein comparison between g-sEV and pg-sEV, 339 differentially expressed proteins were identified. These proteins accounted for 37.3% of the total protein content in g-sEV and 28.9% in pg-sEV. Among the differentially expressed proteins, 209 proteins showed a decrease in abundance after pasteurization (Figure 6B). This suggests that homogenization and high-temperature heating have an impact on the protein content of milk sEVs, primarily resulting in a decrease in protein concentration.

In the comparison between g-sEV and *p*-sEV, 425 differentially expressed proteins were identified. These proteins accounted for 46.7% of the total protein content in g-sEV and 53.5% in *p*-sEV. Among these, the abundance of 293 proteins decreased in *p*-sEV, while the abundance of 132 proteins increased. This indicates that various processing steps, such as homogenization, pasteurization, concentration, and spray drying, further affect the protein content of sEVs (Figure 6C).

### 3.7. Functional Analysis of Differentially Expressed Proteins

Similarly, proteins with significant differences in pairwise comparison samples were selected for GO enrichment analysis and KEGG pathway analysis. In biological processes, differential proteins mainly participate in various positive regulations, such as positive regulation of DNA binding and positive regulation of phagocytosis. The cellular components they are associated with include the perinuclear region of cytoplasm and the extracellular space. The main molecular functions of these proteins include protein homodimerization activity and calcium ion binding, etc. (Figure 7A). The signaling pathways in which these differential proteins are primarily involved include complement and coagulation cascades, ribosomes, and protein processing in the endoplasmic reticulum (Figure 7B).

Differential proteins in g-sEV and *p*-sEV are primarily involved in biological processes such as Arp2/3 complex mediated actin nucleation and acute phase response. The cellular components they are associated with include the cytosol and extracellular space. These differential proteins have molecular functions such as actin filament binding and calcium ion binding (Figure 7C). The signaling pathways in which these differential proteins are primarily involved include endocytosis, ribosomes, and bacterial invasion of epithelial cells (Figure 7D).

## 4. Discussion

Goat milk is rich in nutrients such as carbohydrates, proteins, and other bioactive substances. To extend the shelf life of goat milk products and eliminate pathogenic microorganisms, a series of processing steps, including homogenization, heat treatment, concentration, and spray drying, are commonly employed. Previous studies have shown that these processing steps can affect the nutritional composition of milk, such as denaturation of heat-labile proteins and destruction of certain enzymes during heat treatment [3,16]. Like other dairy emulsions, goat milk also contains a significant amount of sEVs. However, limited research has been conducted on the changes in important components of sEVs, such as miRNA and proteins, after goat milk is processed into milk powder. The extraction method of exosomes used in our study was similar to that reported by González et al. [17]. The results indicate that the use of rennet to remove impurities other than exosomes is an essential step in the extraction of exosomes from goat’s milk with high fat and casein content [18]. Our study revealed that the quantity of sEVs in raw milk did not significantly change after pasteurization, but the quantity of sEVs in milk powder was significantly reduced. The processing steps involved in milk powder production, including homogenization, heat treatment, concentration, and spray drying, led to a loss of sEVs, consistent with the destruction of sEVs in cow milk due to processes such as high-temperature transient treatment, high pressure, and homogenization [19]. Additionally, we found that the purity of *p*-sEV was lower than that of g-sEV and pg-sEV. This lower purity may also be attributed to small extracellular vesicle damage caused by the processing steps, resulting in the leakage of exosomal contents because the protein concentration did not significantly decrease while the quantity of *p*-sEV decreased. Furthermore, we also found that the size of sEVs decreased as the processing steps increase, indicating that larger particles were more susceptible to damage during processing.

A large amount of miRNAs has been found in milk sEVs, which are protected by the phospholipid bilayer. It has been demonstrated that miRNAs in milk sEVs can withstand conditions such as RNase digestion, low pH, and in vitro gastrointestinal digestion [20,21]. However, there is limited research on how different physiological environments and processing techniques affect miRNAs in milk sEVs. There are reports which have shown that processing methods such as high-temperature flash treatment and homogenization can reduce the amount of miRNAs in milk sEVs [22,23,24]. Although pasteurization does not affect the quantity of sEVs, it does affect their overall integrity, appearance, and the content such as miRNAs and proteins [25]. The proportion of miRNAs in all RNAs is also affected, especially in powdered milk sEVs, which decreased by more than 50% (58% vs. 50% vs. 25%). Additionally, the proportion of unannotated RNA in powdered milk significantly increased, further indicating the damage caused by processing. However, we found that the most abundant miRNA species in the three types of sEVs did not show significant differences. For example, the presence of chi-miR-30a-5p, which has been reported to alleviate LPS-induced inflammatory responses in IEC-6 cells, underscores the potential functional relevance of these miRNAs [26]. Chi-let-7b-5p and chi-let-7f-5p were the two most abundant miRNAs in all three types of sEVs, while the top 10 most abundant miRNAs in raw milk and pasteurized milk, although with some differences in ranking, were the same in terms of species. Among the top 10 miRNAs in powdered milk, seven were same as that in the other two types of milk. This suggests that sEVs provide a certain level of protection. Consistent with the findings of Shome et al., our study confirmed that pasteurized milk has a smaller impact on miRNAs in milk sEVs compared to other commercial processing methods [27].

Among the significantly different miRNAs, chi-miR-150, has been found to be highly expressed in goat, human, and bovine milk [28], and its main function is to control B cell differentiation and promote NK cell development. Other downregulated miRNAs in the comparison between g-sEV and *p*-sEV, and g-sEV and pg-sEV are primarily involved in regulating inflammation and development [29,30,31,32], but they are not present in high abundance in goat milk.

Subsequent GO enrichment analysis unveiled that the differential miRNA target genes between g-sEV and pg-sEV, as well as between raw goat milk sEVs and powdered goat milk sEVs, were found to be involved in neural-related processes, such as neuronal migration, brain development, and neuronal projection development, alongside axon guidance signaling pathways. Furthermore, the downregulated miRNA target genes in powdered milk were also associated with pathways such as the ErbB signaling pathway and the MAPK signaling pathway, which are known to play crucial roles in immune response and development [33].

Given the numerous proteins that have been identified in milk, which is sensitive to hard conditions such as heat, the effect of processing on milk proteins has long been a topic of concern. As proteomics has evolved, studies have been conducted to explore the impact of processing on the proteomics of milk [25,34,35,36,37]. Moreover, in recent years, proteomic methods have been employed to investigate the influence of processing on protein modifications [38,39].

Further research is needed to determine whether sEVs can play a protective role for proteins under processing. Like protein change in milk after heat processing [34], we found that pasteurization did not affect the numbers of proteins identified in pg-sEV. However, the identified proteins in *p*-sEV significantly decreased, meaning that homogenization and spray drying could decrease the proteins in sEVs, which was consistent with the findings of protein decrease in spray-dried camel milk powder and bovine milk powder [40,41]. Previous studies have reported that the quantity of exosomes and their contents, such as miRNAs and proteins, are influenced by homogenization under high pressure and shear forces [42]. In contrast, our study demonstrated that spray drying further affects both the quantity of exosomes and their contents.

Similar to miRNA, the protein composition with the highest abundance in the three types of extracellular vesicles is not significantly different. Butyrophilin (BTN1A1), an important functional component in milk, which is associated with lactation, T-cell regulation, and other functions, is the most abundant protein in all three types of extracellular vesicles. Its expression increases in the mammary gland during lactation [43,44]. Other than BTN1A1, both XDH and MFG8 were found to be highly expressed in extracellular vesicles. All these proteins are considered to be highly abundant proteins associated with milk fat globules [45]. Additionally, membrane proteins such as the extracellular vesicle marker proteins CD36 and CD9 are also present in the highly abundant proteins.

Further analysis of differential proteins after processing revealed a significant decrease of some immune-related proteins such as coronin and immunoglobulin J chain, especially in *p*-sEV. Moreover, proteins involved in antimicrobial activity, complement activation, and the lectin pathway are reduced in *p*-sEV. For example, keratin 1 can activate complement and lectin pathways by binding to mannose, thereby activating endothelial cells [46]. This suggests that the processing of dairy products may lead to a reduction in proteins involved in immune-related biological processes and signaling pathways, such as the complement and coagulation cascades, which are activated to prevent pathogen invasion and limit further bleeding [47].

## 5. Conclusions

The results indicate that processing procedures significantly affect the quantity and content of sEVs in goat milk, including inflammatory and immune-related miRNAs and proteins. Furthermore, among the three types of exosomes, the abundance of the top ten miRNAs and proteins largely remains unchanged, suggesting a protective nature of exosomes towards their internal cargo, though this protection is limited. As the processing becomes more extensive and complex, various aspects of sEVs in the milk emulsion are impacted, including a reduction in particle count and size, as well as a decline in the abundance of functional contents such as miRNAs and proteins. Among the three milk-derived sEVs, *p*-sEV exhibited the lowest miRNA quantity and protein abundance. Since miRNAs and proteins in sEVs can be absorbed by human cells and influence metabolic regulation, the findings of this study may provide insights and guidance for dairy processing.

## Figures and Tables

**Figure 1 nutrients-16-04331-f001:**
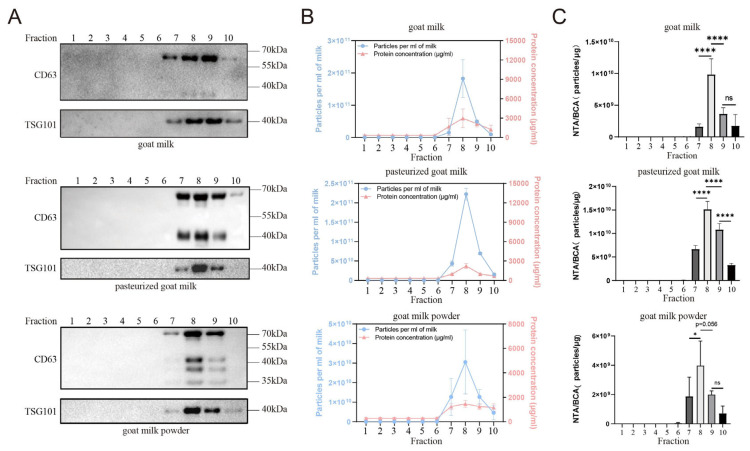
Comparison of exosome content and purity in different fractions of goat dairy products. (**A**) Detection of exosome marker proteins CD63 and TSG101 in different fractions of different goat dairy products by Western blot, (**B**) NTA and BCA were used to detect the content of exosomes (blue) and protein concentration (red) in each fraction of different goat milk products, and (**C**) The purity of exosomes in different fractions of different goat milk products was measured by NTA/BCA. All data are presented as mean ± SD, *p* < 0.05 (*), or *p* < 0.0001 (****), no significance (ns).

**Figure 2 nutrients-16-04331-f002:**
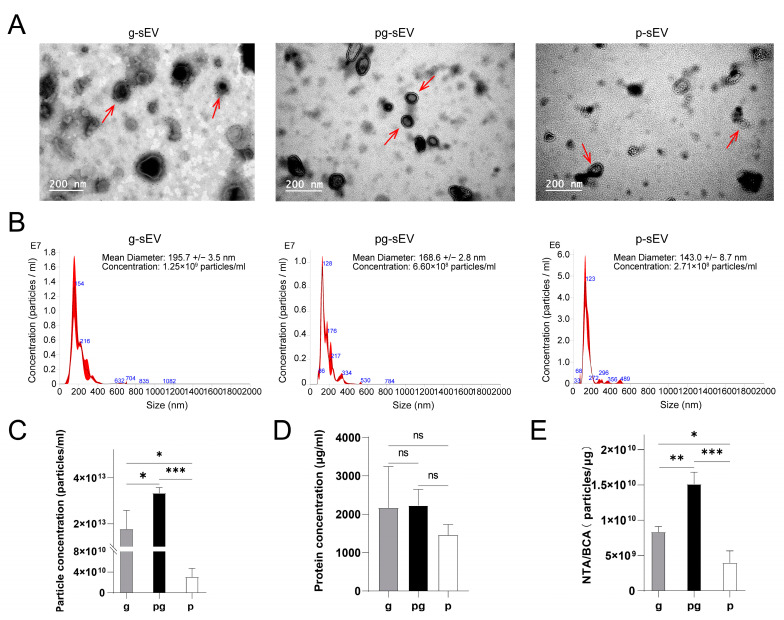
Identification and comparison of milk small extracellular from different goat milk products. (**A**) TEM observation of the shape and size of small extracellular (red arrows) from different goat milk products; the bar—200 nm; (**B**) the diameter distribution of small extracellular in different goat milk products were detected by NTA. (**C**) Comparison of small extracellular content in different goat dairy products; (**D**) comparison of protein concentration in different goat dairy products; (**E**) comparison of the purity of different goat milk products, *p* < 0.05 (*), *p* < 0.01 (**), *p* < 0.001 (***), no significance (ns).

**Figure 3 nutrients-16-04331-f003:**
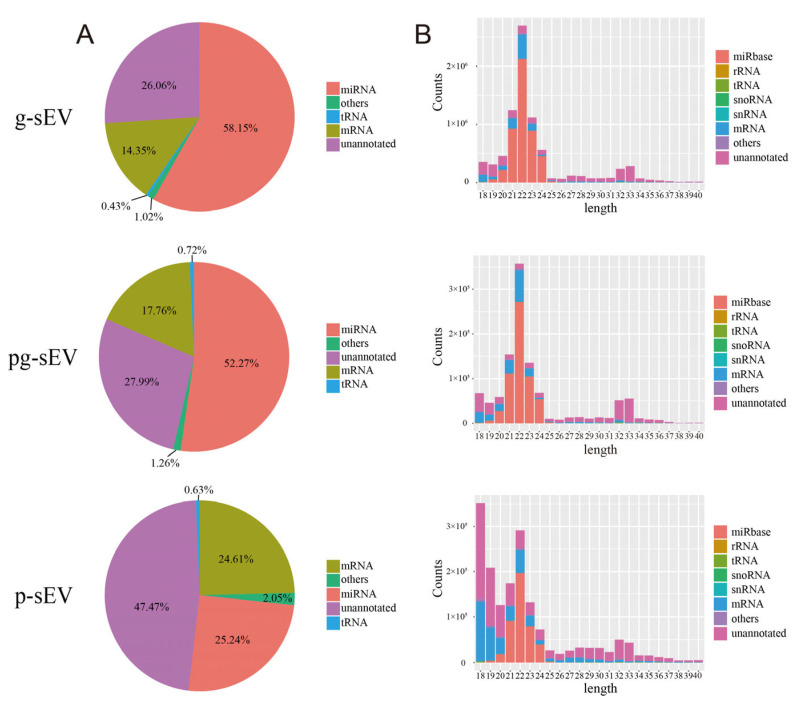
Annotation, classification, and length distribution of small RNAs in milk exosomes of different goat dairy products. (**A**) Comparison of small RNA abundance in exosomes from different dairy products; (**B**) the length distribution of various small RNAs in exosomes of different goat dairy products.

**Figure 4 nutrients-16-04331-f004:**
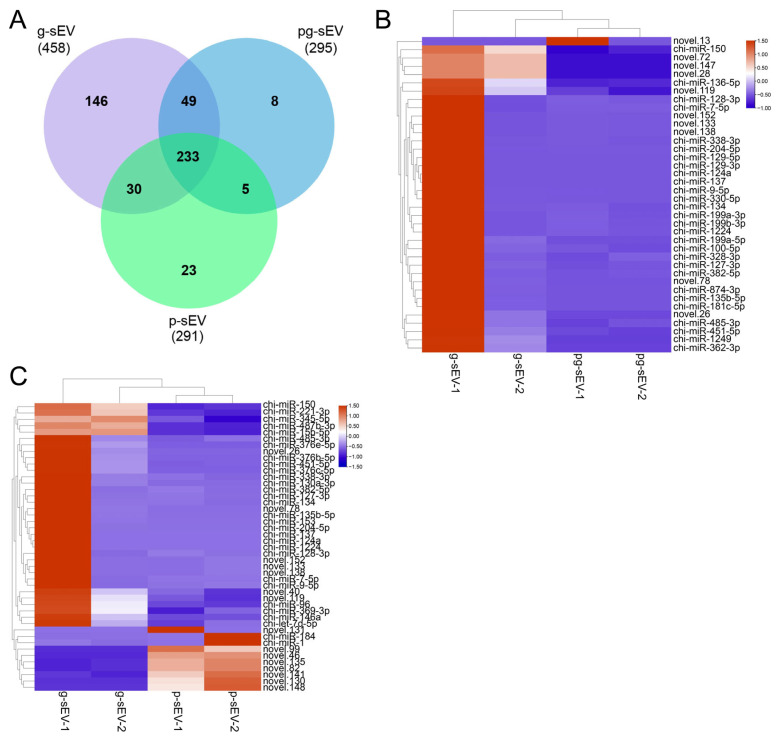
miRNA identification and cluster analysis of differential miRNAs. (**A**) Venn diagram analysis of miRNAs identified in exosomes of different goat milk products; (**B**) heatmap of differential miRNAs in g-sEV and pg-sEV comparison; (**C**) heatmap of differential miRNAs in g-sEV and *p*-sEV comparison.

**Figure 5 nutrients-16-04331-f005:**
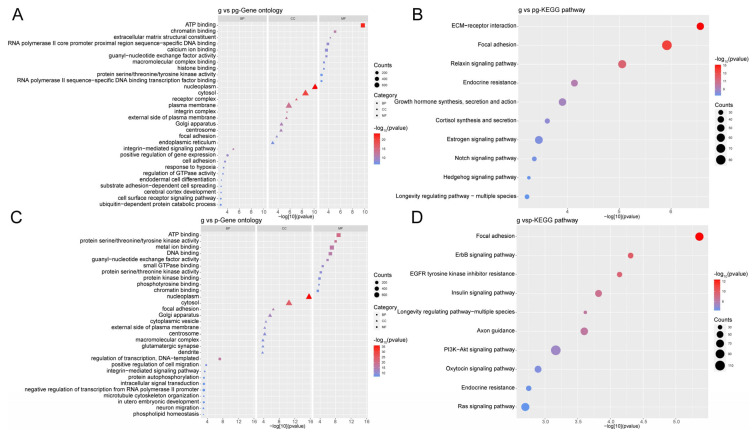
GO and KEGG analysis of the target genes of the top 10 differentially expressed miRNAs. (**A**) GO analysis of target genes of top 10 differentially expressed miRNAs in g-sEV and pg-sEV comparison; (**B**) KEGG analysis of the target genes of the top 10 differentially expressed miRNAs between g-sEV and pg-sEV; (**C**) GO analysis of the target genes of the top 10 differentially expressed miRNAs between g-sEV and *p*-sEv; (**D**) KEGG analysis of target genes of top 10 differentially expressed miRNAs between g-sEV and *p*-sEV.

**Figure 6 nutrients-16-04331-f006:**
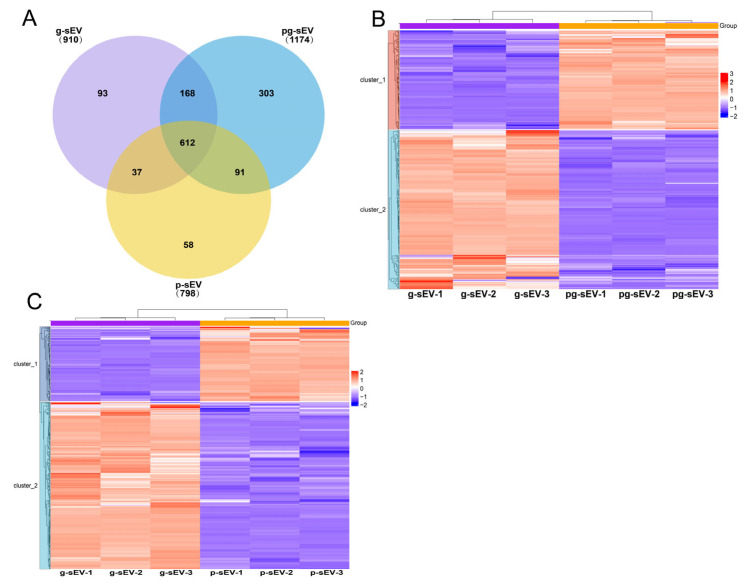
Protein identification and differentially expressed protein cluster analysis. (**A**) Venn diagram was used to analyze the proteins identified in exosomes of different goat milk products. (**B**) Heat map of differentially expressed proteins in g-sEV and pg-sEV comparison; (**C**) heatmap of differentially expressed proteins in g-sEV and *p*-sEV comparison.

**Figure 7 nutrients-16-04331-f007:**
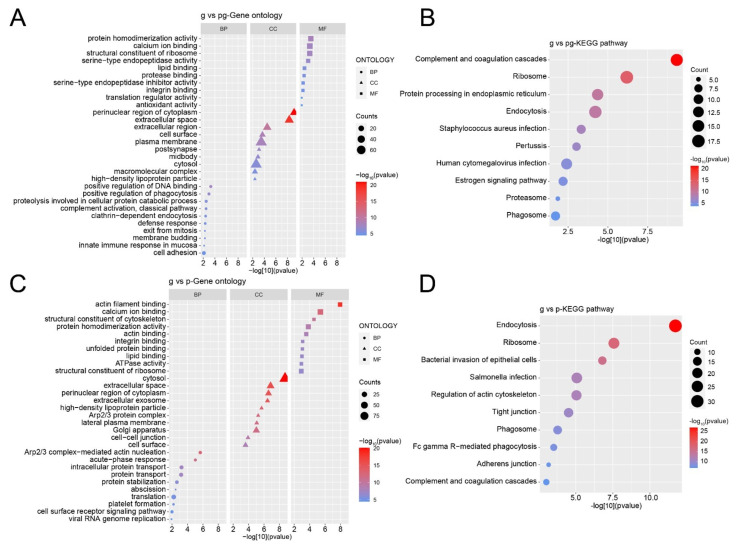
GO and KEGG analysis of differential proteins. (**A**) GO analysis of differential proteins in g-sEV and pg-sEV comparison; (**B**) KEGG analysis of differential proteins between g-sEV and pg-sEv; (**C**) GO analysis of differential proteins in g-sEV and *p*-sEV comparison; (**D**) KEGG analysis of differential proteins in g-sEV and *p*-sEV comparison.

**Table 1 nutrients-16-04331-t001:** Comparison of small extracellular vesicles from different goat milk products.

Parameters	g-sEV	pg-sEV	*p*-sEV
Diameters (nm)	195.7	168.6	143.0
Particle concentration (particles/mL)	1.77 × 10^13^	3.34 × 10^13^ *	3.06 × 10^10^ *
Particles per mL milk (particles)	1.18 × 10^11^	2.2 × 10^11^ *	1.53 × 10^8^ ***
Protein concentration (μg/mL)	2169.47	2231.41 ^ns^	1458.3 ^ns^
NTA/BCA (particles/μg)	8.34 × 10^9^	1.51 × 10^10^ **	3.98 × 10^9^ ***

Legend: Statistical data on the size, particle number, concentration, protein concentration, and purity of three milk-derived sEVs: raw milk small extracellular vesicles (g-sEV), pasteurized milk small extracellular vesicles (pg-sEV), and powdered milk small extracellular vesicles (*p*-sEV). * indicates significant differences compared to g-sEV, *p* < 0.05; *p* < 0.01 (**), *p* < 0.001 (***); ns indicates no significant difference compared to g-sEV.

**Table 2 nutrients-16-04331-t002:** The top 10 miRNAs in the three small extracellular vesicles.

Ordinal	g-sEV	pg-sEV	*p*-sEV
1	chi-let-7b-5p	chi-let-7f-5p	chi-let-7b-5p
2	chi-let-7f-5p	chi-let-7b-5p	chi-let-7f-5p
3	chi-miR-141	chi-miR-16a-5p	chi-miR-148a-3p
4	chi-miR-29a-3p	chi-miR-141	chi-miR-141
5	chi-miR-16a-5p	chi-miR-223-3p	chi-miR-30a-5p
6	chi-miR-148a-3p	chi-miR-30a-5p	chi-miR-16a-5p
7	chi-miR-30a-5p	chi-miR-148a-3p	chi-miR-21-5p
8	chi-miR-26b-5p	chi-miR-146b-5p	chi-miR-29a-3p
9	chi-miR-223-3p	chi-miR-26b-5p	chi-miR-200a
10	chi-miR-146b-5p	chi-miR-29a-3p	chi-miR-103-3p

**Table 3 nutrients-16-04331-t003:** The top ten abundant proteins in the three small extracellular vesicles.

Ordinal	g-sEV	pg-sEV	*p*-sEV
1	BTN1A1	BTN1A1	BTN1A1
2	CD36	H-FABP	LALBA
3	XDH	CD36	H-FABP
4	H-FABP	TSPAN1	CD36
5	MFGE8	CSN2	CD9
6	MUC1	ABCG2	XDH
7	CD9	MFGE8	MUC1
8	GLYCAM1	GLYCAM1	MFGE8
9	SLC34A2	MUC1	SLC34A2
10	ABCG2	SLC34A2	MUC15

## Data Availability

All data supporting the findings of this study are available from the corresponding author, Qiuling Xie, upon reasonable request due to the data are part of an ongoing study.

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
