# Peer review of "Effects of Different Processing on miRNA and Protein in Small Extracellular Vesicles of Goat Dairy Products"

_nutrients, 2024, doi:10.3390/nu16244331_

Round 1
Reviewer 1 Report
Comments and Suggestions for Authors
This manuscript describes the effect of different dairy procedures on several aspects of small extracellular vesicles (EM) from goat milk. Results are very interesting for the dairy research. However, there are several aspects related with the originality of the research and the references to previous works, the way of result presentation and the methodology that raise serious concerns.
First of all, there is a lack of citation of some important papers related to goat milk EM (doi: 10.1021/acs.jafc.4c03212, doi: 10.1016/j.ijbiomac.2024.131698, doi: 10.3389/fbioe.2023.1197780, doi: 10.1021/acs.molpharmaceut.1c00182, doi: 10.1002/smll.202105421, doi: 10.3168/jds.2019-17739) and similar papers related to bovine and human milk EM (doi: 10.1002/fsn3.3749, doi: 10.1038/s41598-023-37310-x). Strikingly, these papers may reduce part of the originality of the present paper. Authors should take special care in order to adequately present/discuss these papers and to properly provide the new input to the field of the present paper.
Regarding methodology, it is extremely relevant that authors indicate how many samples were analysed from each type of EM (raw milk, pasteurized milk and milk powder), how many measurements were performed and which volume of milk was used to perform the analysis. For example, in all the figures, data are presented as mean and SD but no clue of the number of samples analysed or their source or replicas is included. In Fig. 4B y 4C, two different samples are indicated from each type but no indication of the differences is given.
Western blot analysis lacks loading control and positive controls samples (Figure 1A). In addition, an explanation of the multiple bands should be included.
Lines 379-380: “Our study revealed that the quantity of sEVs in raw milk did not significantly change after pasteurization, but the quantity of sEVs in milk powder was significantly reduced”. This statement is not supported in any of the results shown.
Minor points:
-Size of all the Figures should be increased
-Line 228: “1013”
-Line 409: “confirned”
-Line 449: “protiens”
Author Response
Thank you for your careful review and evaluation of the details, cited literature, discussion, and results sections of our manuscript. We have thoroughly considered your suggestions and provided corresponding responses below.
Comments1: First of all, there is a lack of citation of some important papers related to goat milk EM (doi: 10.1021/acs.jafc.4c03212, doi: 10.1016/j.ijbiomac.2024.131698, doi: 10.3389/fbioe.2023.1197780, doi: 10.1021/acs.molpharmaceut.1c00182, doi: 10.1002/smll.202105421, doi: 10.3168/jds.2019-17739) and similar papers related to bovine and human milk EM (doi: 10.1002/fsn3.3749, doi: 10.1038/s41598-023-37310-x). Strikingly, these papers may reduce part of the originality of the present paper. Authors should take special care in order to adequately present/discuss these papers and to properly provide the new input to the field of the present paper.
Response1: Thank you very much for your constructive suggestions regarding the references in our manuscript. We find them very helpful, and we have cited and modified some of them in our article, which significantly enhances its completeness.
The following references were included:
doi: 10.1016/j.ijbiomac.2024.131698, lines 425-428
doi: 10.3389/fbioe.2023.1197780, lines 397-399
doi: 10.1002/fsn3.3749, lines 462-466
Some of these references focus on the functions of sEVs, while others study methods for sEVs research. The last reference discusses the impact of processing on milk sEVs. Our study, however, examines the effects of processing on the sEVs of goat milk, specifically on the commonly available products of pasteurized milk and milk powder, which is currently an unexplored area.
As for the other references, we reviewed them but found that their focus differs from that of our research, so we did not include them further. Nonetheless, they were still very beneficial to us.
Comments2: Regarding methodology, it is extremely relevant that authors indicate how many samples were analysed from each type of EM (raw milk, pasteurized milk and milk powder), how many measurements were performed and which volume of milk was used to perform the analysis. For example, in all the figures, data are presented as mean and SD but no clue of the number of samples analysed or their source or replicas is included. In Fig. 4B y 4C, two different samples are indicated from each type but no indication of the differences is given.
Response2: Thank you for pointing out this issue. We mentioned the sample volume used in the original manuscript, specifically in lines 81-82, where we state that 200 ml of sample is used for each extraction. Regarding the sample source, due to the unique challenges in obtaining and processing the samples, it is difficult to acquire samples from the same batch, especially for milk powder. In fact, we compared commercial goat milk products produced by the same manufacturer, which are available on the market, across three different batches, with each batch extracted and tested separately. The raw milk was supplied by Ausnutria Hyproca Nutrition Ltd. in Hunan, China, sourced from the same breed of goat (Saanen goats). In the comparison shown in Figure 4, for instance in Figure 4B, the g-sEV come from different batches of samples, but the same extraction method was used, and the same applies to pg-sEV.
Comments3: Western blot analysis lacks loading control and positive controls samples (Figure 1A). In addition, an explanation of the multiple bands should be included.
Response3: Thank you very much for highlighting this issue. We would like to clarify that CD63 and TSG101 are recognized marker proteins for sEVs. We measured the concentrations of these two proteins in different fractions after chromatography and confirmed that sEVs concentration primarily appeared in fraction 8, which helped us establish the sEVs extraction protocol.
Regarding the control, current literature on the detection of CD63 and TSG101 in exosomes typically does not include a positive control. If the emulsion itself were used as a control, the protein concentration might be too low to be detected because exosome extraction is an enrichment process.
Additionally, there are multiple bands for CD63, as it encodes a tetraspanin protein with a molecular weight range of 30 to 60 kDa (doi:10.1007/PL00000933). This protein is also subject to glycosylation and various interactions with cell surface proteins. We observed that, compared to raw milk, processing resulted in an increase in the number of CD63 bands, with the milk powder showing the most bands. This suggests that as the degree of processing increases, CD63 may undergo degradation.
Comments4: Lines 379-380: “Our study revealed that the quantity of sEVs in raw milk did not significantly change after pasteurization, but the quantity of sEVs in milk powder was significantly reduced”. This statement is not supported in any of the results shown.
Response4: As shown in Figure 2C, we compared the particle concentrations of three groups: p-sEV, g-sEV, and pg-sEV. The particle concentration of p-sEV was significantly lower than that of g-sEV and pg-sEV, while the concentration of g-sEV was lower than that of pg-sEV.
Comments5: -Size of all the Figures should be increased
Response5: Thank you for your careful review of our manuscript. We have made corrections to the errors noted, as indicated in lines 241, 433, and 457.
Reviewer 2 Report
Comments and Suggestions for Authors
The article deals with an interesting problem related to the quality and quantitative composition of goat's milk before and after its processes. Mirna and Protein in Small Extracellular Vesicles of Goat Dairy Products have not been studied so far.
The manuscript has a huge number of experimental and bioinformatic data, proteomics, and proving certain proteins with precision analytical techniques.
I think there is no need for any changes to the text.
The figures must be enlarged in order to see the designations in them.
I have the following questions:
1. How do you explain the lack of expression of protein related to shock conditions? How is drying by lyophilization of the protein profile in terms of stress factors?
2. Do you notice any difference in the proteome profile associated with transport proteins?
3. The differences in the lipid profile have not been discussed anywhere, and this aspect is important given the composition of the vesicles. What does the lipid bite of the vesicles, what fats, and how are the different conditions of the components?
Author Response
Thank you for your recognition of our manuscript and for your discussions regarding the proteomics aspect. We have carefully reviewed your comments and provided our corresponding responses below.
Comments1: 1. How do you explain the lack of expression of protein related to shock conditions? How is drying by lyophilization of the protein profile in terms of stress factors?
Response1: Thank you for raising the questions and discussions regarding our proteomics data. In fact, there are numerous proteins associated with shock in the overall proteomic data. Among the differentially expressed proteins, several are related to shock, such as BPI, ARFGEF1, IQGAP2, PPP2R1A, ADA, GPRC5A, and APOF found in g-sEV and p-sEV. Additionally, proteins such as ABCG2, C1Q, ENTDP1, ELANE, FCN1, and ADA in the comparison between g-sEV and pg-sEV have also been reported to be associated with shock conditions.
The milk powder processing primarily involves spray drying, homogenization, and high temperatures, and does not include freeze-drying steps. Therefore, this study does not address the changes that might occur in proteins due to freeze-drying.
Comments2: Do you notice any difference in the proteome profile associated with transport proteins
Response2: Thank you for your questions and discussions regarding the proteomics in our manuscript. In fact, there are several proteins associated with transport in the overall proteomic data. Among the differentially expressed proteins, there are transport-related proteins, such as ELMOD2, ARFGEF1, TMED9, MYO6, and SEC22B in the comparison between g-sEV and p-sEV. Additionally, in the comparison of g-sEV and pg-sEV, proteins such as ARFGEF1, ABCG2, SPART, ENTPD1, and CDC42SE1 are also included.
Comments3: The differences in the lipid profile have not been discussed anywhere, and this aspect is important given the composition of the vesicles. What does the lipid bite of the vesicles, what fats, and how are the different conditions of the components
Response3: Thank you for the research insights you provided regarding our manuscript. The differences in lipidomics are indeed very important; however, in this paper, we focused on miRNA and proteomics because these two types of substances often play critical physiological roles. We are currently conducting research on exosomal lipidomics.
Reviewer 3 Report
Comments and Suggestions for Authors
MS ID: nutrients-3274269
In the article entitled "Effects of Different Processing on miRNA and Protein in Small Extracellular Vesicles of Goat Dairy Products", the authors address the SEVs collected from raw goat milk (g-sEV), pasteurised goat milk (pg-sEV) and goat milk powder (p-sEV). Then, SEVs were identified and compared by NTA, Western blot and TEM, RNA was subjected to high-throughput sequencing and peptide fragments were analysed by mass spectrometry. Finally, GO and KEGG pathway analyses were performed on the obtained data. The sequencing results showed that all three types of small extracellular vesicles were miRNA-rich, and no significant differences were observed in the most abundant SEV species. Compared with g-sEVs, 3,938 and 4,645 differentially expressed miRNAs were found in pg-sEVs and p-sEVs, respectively, most of which (3,837 and 3,635) were downregulated. The authors note that these differentially expressed miRNAs affect biological processes or pathways, such as neurodevelopment, embryonic development, and transcription. Proteomic analysis revealed that there were 339 differentially expressed proteins between g-sEV and pg-sEV, of which 209 were downregulated. However, no significant differences were observed in the most abundant protein species between the three SEV types.
The paper is interesting and in line with the aims of the journal, but needs major revisions. below are suggestions for the authors, line by line:
Lines 37-38: Goat's milk is one of the main dairy foods and is believed to have a shorter gastric emptying time and better digestibility than other types of milk [1]. I disagree, donkey milk also has similar characteristics.
Line 46: sEVs are small. Rewrite as “Small Extracellular vesicles (sEVs)”
Lines 61-68: Although there is extensive research on human and bovine milk sEVs, limited studies have been conducted on goat milk sEVs, particularly regarding the effects of processing on their composition. Given the biological functions and contents of sEVs, it is worthwhile to explore the components of goat milk sEVs and their processed products. In this study, we employed sucrose cushion centrifugation and size exclusion chromatography to extract sEVs from raw goat milk, pasteurized goat milk, and goat milk powder. Subsequently, miRNA sequencing and proteomics were utilized to analyze and compare the sEVs from the three types of milk, aiming to investigate the impact of processing on goat milk sEVs, as well as their miRNA and protein contents. Clearly describe the aim of the study.
Lines 82-83: Subsequently, the skimmed raw goat milk, pasteurized milk and goat milk powder were pretreated with chymosin (TCI). Verify and better describe the use of chymosin for the production of goat milk powder.
Lines 138-140: Small RNA library construction and miRNA sequencing were performed by SHBio Biotechnology Co., Ltd. The extracted RNA was purified using RNA Clean XP Kit (Beckman Coulter) and RNase-Free DNase Set Kit (Qiagen). Check the sentence and insert the reference website.
Lines 152-153: The corrected p-value (q-value) and fold change were used to identify differentially expressed miRNAs for further analysis. The q-value is a measure used in multiple hypothesis testing to control the false discovery rate (FDR). Explanation.
Line 158: vesicle protein. Perhaps vesicle proteins.
Lines 183-185: The statistical significance of enrichment was calculated using the Benjamini-Hochberg p-value adjustment method, and the enrichment results were visualized using the R package ggplot2. A "statistical analysis" paragraph needs to be inserted.
Lines 227-228: In addition, the particle concentration of p-sEV (3.06×1010 particles) was significantly lower than that of g-sEV (1.77×1013 particles) and pg-sEV (3.34×1013 particles) (Figure 2C, Table 1). These results are also reported in Table 1, but no significant differences between the parameters are indicated.
Lines 281-286: Figure 4. I think the figure needs to be enlarged.
Lines 317-319: Statistical comparison showed that 910, 1174, and 798 proteins were identified in g-sEV, pg-sEV, and p-sEV, respectively. Describe, see also the suggested comment inserted in lines 183-185.
Lines 373-375: Figure 7. I think the figure needs to be enlarged.
Lines 456-463: 5. Conclusion.
The above results indicate that processing procedures mainly affect the number and content of sEVs in goat milk, including inflammatory and immune-related miRNAs and proteins. The more processing treatments, the greater the loss of miRNA and proteins in sEVs. On the other hand, the most abundant miRNA and protein species in the three sEVs were not found to be significantly different, indicating that sEVs have a certain protective effect. Because the miRNAs and proteins in sEVs can be taken up by human cells and affect metabolic regulation [45,46], the results of this study may provide insight and help in milk processing.
The conclusions need to be rewritten, taking into account what is stated in the objectives of the study. Although these conclusions appear to be a discussion, they should not include references.
Comments on the Quality of English Language
In my opinion, a thorough revision of the English language is necessary.
Author Response
Thank you for your careful review and evaluation of the details, cited literature, discussion, and results sections of our manuscript. We have thoroughly considered your suggestions and provided corresponding responses below.
Comments1: Lines 37-38: Goat's milk is one of the main dairy foods and is believed to have a shorter gastric emptying time and better digestibility than other types of milk [1]. I disagree, donkey milk also has similar characteristics.
Response1: Thank you for your careful review of our manuscript and the questions raised. We further consulted additional literature and found that, as described in the reference (https://doi.org/10.1093/bja/aeu338), the gastric emptying time of liquids is related to the total caloric content, and differences in fat content significantly influence the total calorie count. Therefore, the varying fat content in milk from different sources may affect gastric emptying and digestibility. As a result, we revised our statement to compare only sheep milk with cow milk. Please see lines 39-40 for the changes.
Comments2: Line 46: sEVs are small. Rewrite as “Small Extracellular vesicles (sEVs)”
Response2: Thank you for your careful review of our manuscript. We have made revisions in the original text. Please see line 48 for the changes.
Comments3: Lines 61-68: Although there is extensive research on human and bovine milk sEVs, limited studies have been conducted on goat milk sEVs, particularly regarding the effects of processing on their composition. Given the biological functions and contents of sEVs, it is worthwhile to explore the components of goat milk sEVs and their processed products. In this study, we employed sucrose cushion centrifugation and size exclusion chromatography to extract sEVs from raw goat milk, pasteurized goat milk, and goat milk powder. Subsequently, miRNA sequencing and proteomics were utilized to analyze and compare the sEVs from the three types of milk, aiming to investigate the impact of processing on goat milk sEVs, as well as their miRNA and protein contents. Clearly describe the aim of the study.
Response3: Thank you for your careful review of our manuscript. We have made revisions to the original text to clarify the research objectives. The modifications can be found in lines 72-73.
Comments4: Lines 82-83: Subsequently, the skimmed raw goat milk, pasteurized milk and goat milk powder were pretreated with chymosin (TCI). Verify and better describe the use of chymosin for the production of goat milk powder.
Response4: Thank you for your careful review of our manuscript and for pointing out the issues with the experimental methods. We have added more detailed information about the rennet and treatment methods. Specifically, we added rennet to the skim milk at a concentration of 0.0002 g/mL, incubated at 37°C for 1 hour, and then centrifuged at 10,000 g for 30 minutes (4°C) to collect the supernatant. Please see lines 88-89 for the details.
Comments5: Lines 138-140: Small RNA library construction and miRNA sequencing were performed by SHBio Biotechnology Co., Ltd. The extracted RNA was purified using RNA Clean XP Kit (Beckman Coulter) and RNase-Free DNase Set Kit (Qiagen). Check the sentence and insert the reference website.
Response5: Thank you for your careful review of our manuscript and for pointing out the issues with the experimental methods. We have rewritten this sentence to specify the companies involved, the different functions of the kits, and the relevant websites. The modifications can be found in lines 143-146.
Comments6: The corrected p-value (q-value) and fold change were used to identify differentially expressed miRNAs for further analysis. The q-value is a measure used in multiple hypothesis testing to control the false discovery rate (FDR). Explanation.
Response6: Thank you for your careful review of our manuscript and for pointing out the issues with the experimental methods. Upon review, we found that this passage contained some ambiguities, so we have rewritten it and provided explanations. The modifications can be found in lines 173-198.
Comments7: Line 158: vesicle protein. Perhaps vesicle proteins.
Response7: Thank you for your careful review of our manuscript. We have made the necessary revisions.
Comments8: Lines 183-185: The statistical significance of enrichment was calculated using the Benjamini-Hochberg p-value adjustment method, and the enrichment results were visualized using the R package ggplot2. A "statistical analysis" paragraph needs to be inserted.
Response8: Thank you for your comments on the data analysis section of our manuscript. We have provided detailed explanations of our methods for both the miRNA sequencing results and the mass spectrometry results in section 2.8 (lines 173-198). We have also included a thorough explanation of the Benjamini-Hochberg p-value adjustment method.
Comments9: Lines 227-228: In addition, the particle concentration of p-sEV (3.06×1010 particles) was significantly lower than that of g-sEV (1.77×1013 particles) and pg-sEV (3.34×1013 particles) (Figure 2C, Table 1). These results are also reported in Table 1, but no significant differences between the parameters are indicated.
Response9: Thank you for pointing out the issues with our data on exosome particle counts. As shown in Figure 2C, we compared the particle concentrations of three groups: p-sEV, g-sEV, and pg-sEV, and found significant differences. Specifically, the particle concentration of p-sEV was significantly lower than that of g-sEV and pg-sEV, while the concentration of g-sEV was lower than that of pg-sEV.
Comments10: Lines 281-286: Figure 4. I think the figure needs to be enlarged.
Response10: Thank you for raising the issue regarding the quality of the images in our manuscript. The original images are indeed very clear (PPI=300); however, due to formatting constraints, we reduced their size. We have also uploaded the original high-resolution images to the system. Additionally, we included the top ten differentially expressed miRNAs in Table 2 to present the data more clearly to the readers.
Comments11: Lines 317-319: Statistical comparison showed that 910, 1174, and 798 proteins were identified in g-sEV, pg-sEV, and p-sEV, respectively. Describe, see also the suggested comment inserted in lines 183-185.
Response11: Thank you for your comments on the data analysis section of our manuscript. We have provided detailed explanations of our methods for both the miRNA sequencing results and the mass spectrometry results in section 2.8 (lines 173-198). We have also included a thorough explanation of the Benjamini-Hochberg p-value adjustment method.
Comments12: Lines 373-375: Figure 7. I think the figure needs to be enlarged.
Response12: Thank you for raising the issue regarding the quality of the images in our manuscript. The original images are indeed very clear (PPI=300); however, due to formatting constraints, we reduced their size. We have also uploaded the original high-resolution images to the system. Additionally, we included the top ten differentially expressed miRNAs in Table 2 to present the data more clearly to the readers.
Comments13: The above results indicate that processing procedures mainly affect the number and content of sEVs in goat milk, including inflammatory and immune-related miRNAs and proteins. The more processing treatments, the greater the loss of miRNA and proteins in sEVs. On the other hand, the most abundant miRNA and protein species in the three sEVs were not found to be significantly different, indicating that sEVs have a certain protective effect. Because the miRNAs and proteins in sEVs can be taken up by human cells and affect metabolic regulation [45,46], the results of this study may provide insight and help in milk processing.
The conclusions need to be rewritten, taking into account what is stated in the objectives of the study. Although these conclusions appear to be a discussion, they should not include references.
Response13: Thank you very much for your careful review of our manuscript and for your valuable suggestions regarding the writing of the conclusion section. Based on your feedback and our own reflections, we have rewritten the conclusion, as shown in lines 488-499.
Round 2
Reviewer 1 Report
Comments and Suggestions for Authors
Authors adequately addressed some of the questions raised by this reviewer. However, there are still some points that should be improved:
-In the author´s response, they indicate that “we compared commercial goat milk products produced by the same manufacturer, which are available on the market, across three different batches, with each batch extracted and tested separately…. In the comparison shown in Figure 4, for instance in Figure 4B, the g-sEV come from different batches of samples, but the same extraction method was used, and the same applies to pg-sEV”. These explanations should be included in the text.
-In the author´s response, they comment that “Additionally, there are multiple bands for CD63, as it encodes a tetraspanin protein with a molecular weight range of 30 to 60 kDa (doi:10.1007/PL00000933). This protein is also subject to glycosylation and various interactions with cell surface proteins. We observed that, compared to raw milk, processing resulted in an increase in the number of CD63 bands, with the milk powder showing the most bands. This suggests that as the degree of processing increases, CD63 may undergo degradation”. These comments should be included in the text.
-Regarding lines 392-393: “Our study revealed that the quantity of sEVs in raw milk did not significantly change after pasteurization,…” in the author´´s response they indicate that “As shown in Figure 2C,… The particle concentration of p-sEV was significantly lower than that of g-sEV and pg-sEV, while the concentration of g-sEV was lower than that of pg-sEV.” That means that pasteurization (pg-sEV) increases the particle concentration. This should be corrected.
Author Response
Comment1:-In the author´s response, they indicate that “we compared commercial goat milk products produced by the same manufacturer, which are available on the market, across three different batches, with each batch extracted and tested separately…. In the comparison shown in Figure 4, for instance in Figure 4B, the g-sEV come from different batches of samples, but the same extraction method was used, and the same applies to pg-sEV”. These explanations should be included in the text.
Response1:Thank you very much for your valuable suggestions regarding the additions and modifications to our article. We have incorporated the changes in the Materials and Methods section, which can be found on lines 75-76.
Comment2:-In the author´s response, they comment that “Additionally, there are multiple bands for CD63, as it encodes a tetraspanin protein with a molecular weight range of 30 to 60 kDa (doi:10.1007/PL00000933). This protein is also subject to glycosylation and various interactions with cell surface proteins. We observed that, compared to raw milk, processing resulted in an increase in the number of CD63 bands, with the milk powder showing the most bands. This suggests that as the degree of processing increases, CD63 may undergo degradation”. These comments should be included in the text.
Response2:Thank you for pointing out the areas in our article that should be modified and added. We have made the additions at the appropriate locations within the results section, which you can find on lines 200-204.
Comment3:-Regarding lines 392-393: “Our study revealed that the quantity of sEVs in raw milk did not significantly change after pasteurization,…” in the author´´s response they indicate that “As shown in Figure 2C,… The particle concentration of p-sEV was significantly lower than that of g-sEV and pg-sEV, while the concentration of g-sEV was lower than that of pg-sEV.” That means that pasteurization (pg-sEV) increases the particle concentration. This should be corrected.
Response3:Thank you for your thorough review of our article and for your valuable suggestions regarding the discussion section. We have revised the relevant discussions accordingly, and you can find the changes on lines 399-401.
Reviewer 3 Report
Comments and Suggestions for Authors
The authors have done an excellent job in revising the paper in response to the referee's comments. I have just a few minor suggestions, which I will outline below.
Comments1: Lines 37-38: Goat's milk is one of the main dairy foods and is believed to have a shorter gastric emptying time and better digestibility than other types of milk [1]. I disagree, donkey milk also has similar characteristics.
I invite the authors to consider these two bibliographic references, available in open access on the web.
1) Sarti L, Martini M, Brajon G, Barni S, Salari F, Altomonte I, Ragona G, Mori F, Pucci N, Muscas G, Belli F. Donkey’s Milk in the Management of Children with Cow’s Milk protein allergy: Nutritional and hygienic aspects. Italian Journal of Pediatrics. 2019 Dec;45:1-9.
An extract from the introduction
Donkey’s milk (DM) has recently received growing interest as has been reported to be an adequate alternative for children with CMPA and CM-FPIES, mainly due to its nutritional similarities with human milk [23] and excellent palatability and tolerability [24,25,26,27,28,29], unlike the milk of other species, such as goat’s and sheep’s milk, which can lead to cross-reactivity between their proteins and CM proteins [17, 30, 31]. In fact, DM shows a protein fraction more similar to HM than CM, in addition to which, the primary structure of DM’s caseins presents significant differences compared to other species, and it is always more closely related with HM counterparts [18, 32,33,34]. This may contribute towards explaining the less allergenic properties of DM and its greater digestibility [35]. Furthermore, the high lactose content of DM confers good palatability.
2) Lajnaf R, Feki S, Ameur SB, Attia H, Kammoun T, Ayadi MA, Masmoudi H. Cow's milk alternatives for children with cow's milk protein allergy-Review of health benefits and risks of allergic reaction. International Dairy Journal. 2023 Jun 1;141:105624..
Cow's milk protein allergy (CMPA) is considered as the most common food allergy in early life and may cause anaphylaxis reactions in severe cases. This review summarises recent findings in CMPA studies, especially regarding the main relevant cow's milk substitutes such as hydrolysed and plant-based (soy and rice) formulas in addition to other mammalian milk types (goat, sheep, donkey, mare and camel) to reduce allergy risks for children. Extensively hydrolysed cow's milk formulas are mainly used as an alternative for children with CMPA, despite their poor palatability. Goat's and sheep's milk and soy-based formulas are not recommended because of their high cross-reactivity with cow's milk proteins. On the contrary, equine's and camel's milk proteins are suggested as suitable alternative solutions due to their low sequence identity levels with cow's milk proteins. Nonetheless, further research needs to confirm the usefulness of these milk types as a solution in paediatric CMPA.
Comments9: Lines 227-228: In addition, the particle concentration of p-sEV (3.06×1010 particles) was significantly lower than that of g-sEV (1.77×1013 particles) and pg-sEV (3.34×1013 particles) (Figure 2C, Table 1). These results are also reported in Table 1, but no significant differences between the parameters are indicated.
Response9: Thank you for pointing out the issues with our data on exosome particle counts. As shown in Figure 2C, we compared the particle concentrations of three groups: p-sEV, g-sEV, and pg-sEV, and found significant differences. Specifically, the particle concentration of p-sEV was significantly lower than that of g-sEV and pg-sEV, while the concentration of g-sEV was lower than that of pg-sEV
My suggestion for the future is to indicate the superscripts of significance in the table (Table 1 in this case) and in the histograms (Fig. 2C), as the scale of the different concentrations (particle concentration (particles/mL)) on the y-axis is more difficult for the reader to interpret, even if it is correct.
Author Response
Comment1:Comments1: Lines 37-38: Goat's milk is one of the main dairy foods and is believed to have a shorter gastric emptying time and better digestibility than other types of milk [1]. I disagree, donkey milk also has similar characteristics.
I invite the authors to consider these two bibliographic references, available in open access on the web.
1) Sarti L, Martini M, Brajon G, Barni S, Salari F, Altomonte I, Ragona G, Mori F, Pucci N, Muscas G, Belli F. Donkey’s Milk in the Management of Children with Cow’s Milk protein allergy: Nutritional and hygienic aspects. Italian Journal of Pediatrics. 2019 Dec;45:1-9.
An extract from the introduction
Donkey’s milk (DM) has recently received growing interest as has been reported to be an adequate alternative for children with CMPA and CM-FPIES, mainly due to its nutritional similarities with human milk [23] and excellent palatability and tolerability [24,25,26,27,28,29], unlike the milk of other species, such as goat’s and sheep’s milk, which can lead to cross-reactivity between their proteins and CM proteins [17, 30, 31]. In fact, DM shows a protein fraction more similar to HM than CM, in addition to which, the primary structure of DM’s caseins presents significant differences compared to other species, and it is always more closely related with HM counterparts [18, 32,33,34]. This may contribute towards explaining the less allergenic properties of DM and its greater digestibility [35]. Furthermore, the high lactose content of DM confers good palatability.
2) Lajnaf R, Feki S, Ameur SB, Attia H, Kammoun T, Ayadi MA, Masmoudi H. Cow's milk alternatives for children with cow's milk protein allergy-Review of health benefits and risks of allergic reaction. International Dairy Journal. 2023 Jun 1;141:105624..
Cow's milk protein allergy (CMPA) is considered as the most common food allergy in early life and may cause anaphylaxis reactions in severe cases. This review summarises recent findings in CMPA studies, especially regarding the main relevant cow's milk substitutes such as hydrolysed and plant-based (soy and rice) formulas in addition to other mammalian milk types (goat, sheep, donkey, mare and camel) to reduce allergy risks for children. Extensively hydrolysed cow's milk formulas are mainly used as an alternative for children with CMPA, despite their poor palatability. Goat's and sheep's milk and soy-based formulas are not recommended because of their high cross-reactivity with cow's milk proteins. On the contrary, equine's and camel's milk proteins are suggested as suitable alternative solutions due to their low sequence identity levels with cow's milk proteins. Nonetheless, further research needs to confirm the usefulness of these milk types as a solution in paediatric CMPA.
Response1:Thank you very much for your insightful corrections and discussions on the characteristics of goat milk compared to other animal milks in our article, as well as your suggestions for additional references to cite. After carefully studying the two references you provided, we noticed that their discussions are not closely related to the focus of our manuscript, which is goat milk. Therefore, we have not included further citations from these two references. However, they have still provided us with ideas for further research.
Comment2:
Comments9: Lines 227-228: In addition, the particle concentration of p-sEV (3.06×1010 particles) was significantly lower than that of g-sEV (1.77×1013 particles) and pg-sEV (3.34×1013 particles) (Figure 2C, Table 1). These results are also reported in Table 1, but no significant differences between the parameters are indicated.
Response9: Thank you for pointing out the issues with our data on exosome particle counts. As shown in Figure 2C, we compared the particle concentrations of three groups: p-sEV, g-sEV, and pg-sEV, and found significant differences. Specifically, the particle concentration of p-sEV was significantly lower than that of g-sEV and pg-sEV, while the concentration of g-sEV was lower than that of pg-sEV
My suggestion for the future is to indicate the superscripts of significance in the table (Table 1 in this case) and in the histograms (Fig. 2C), as the scale of the different concentrations (particle concentration (particles/mL)) on the y-axis is more difficult for the reader to interpret, even if it is correct.
Response2:Thank you for your suggestions on further refining our charts and tables. We have marked the statistically significant differences in the table data using asterisks (*) and added notes below the table. Regarding your suggestions for improving the figures, we have already made comparisons between the components in the bar chart of Figure 2C and annotated them with horizontal lines. The changes can be found on lines 240-254.